

# Impact of precipitation levels on vegetation in ecologically fragile karst areas in the Guangxi (China) karst region

Mingzhi Li[1], Ying Xie[2], Yanli Chen[2], Yue Zhang[3] and Weihua Mo[2]

[1] Baise Meteorological Bureau, Baise, China
[2] Guangxi Institute of Meteorological Sciences, Nanning, China
[3] College of Resources and Environmental Sciences, China Agricultural University, Beijing, China

## ABSTRACT

To investigate the distribution pattern of regional rainstorm disasters and their impact on vegetation in karst regions of Guangxi, two vegetation parameters, fractional vegetation cover (FVC) and net primary productivity (NPP), are selected to analyze the spatial response characteristics and forest species from five rainfall levels: moderate rainfall, heavy rain, rainstorm, heavy rainstorm and extremely heavy rainstorm). Normalized Difference Vegetation Index (NDVI), fractional vegetation cover (FVC), and net primary productivity (NPP) are used to analyze the spatial response characteristics of different vegetation remote sensing parameters. The results show that: (1) The effects of extremely heavy rainfall on the NDVI, FVC and NPP of vegetation are significantly greater than those of other types of rainfall; (2) The southwestern and central parts are the concentration areas of high negative impacts of extremely heavy rainfall and heavy rainfall on the remote sensing indices of vegetation; (3) Different levels of rainfall have a great negative effect on NDVI and FVC in economic and broadleaf forests, while eucalyptus forests have a less effect. The results indicate that vegetation protection should be carried out in a concentrated manner based on geographical and species-specific differences, especially in areas with high incidence of extremely heavy rainfall and regions dominated by economical value and vegetation types. This study can provide a scientific basis for improving the management of rocky desertification and assessing the impact of rainstorm disaster on vegetation in karst regions of Guangxi.

# INTRODUCTION

Karst is one of the four major ecologically fragile regions in China. Karst landforms are widely distributed in Guangxi, characterized by prominent rocky desertification landscape (*Chen et al., 2018*) and shallow soil layers. In this unique habitat, vegetation is weak for meteorological disasters (*Xu, Hu & Wang, 2012*). In the context of global warming, meteorological disasters occur and develop at a high frequency and intensity (*Zhai & Liu, 2012*; *Lim & Kim, 2023*; *Pandey, Tiwari & Mishra, 2022*), posing a great threat to ecological

Corresponding author
Yanli Chen, yzch208@163.com

environment protection. Rainstorms are one of the most significant meteorological disasters in karst areas (*Huang et al., 2015*; *Cahyadi et al., 2021*). Assessing their impact on these regions is crucial for controlling rocky desertification, protecting and restoring vegetation, and managing the ecological environment.

Many remote sensing vegetation indexes have been developed, and NDVI is the most widely used one (*Khoroshev, Kalmykova & Dusaeva, 2023*; *Mazengo et al., 2023*). FVC refers to the percentage of the vertical projection area of the above-ground part of surface vegetation per unit area to the total statistical area (*Gitelson et al., 2002*). NPP is the organic matter accumulated by green plants per unit area and time, that is, the deduction of respiration consumption of plants from organic carbon fixed by photosynthesis (*Pu, Fang & Guo, 2001*). As an explicit spatial indicator (*Donmez et al., 2024*), it has been proven to be a highly effective indicator of plant community growth and its ecological quality (*Qian et al., 2020*). Meteorological conditions are an indispensable and important factors affecting vegetation growth (*Uffia et al., 2021*; *Tian et al., 2024*; *Nabizada et al., 2023*). Many studies have been conducted using remotely sensed vegetation indices and meteorological data, mainly including the average and non-average (meteorological disasters) impact of climatic factors (meteorological hazards). In terms of the average state, it is generally believed that temperature has significant impact on vegetation in temperate and cold regions, while precipitation has a significant impact on vegetation growth in regions with distinct dry and wet seasons or in arid and semi-arid areas. For example, in temperate grassland areas, for every 1 °C increase in temperature, the average NDVI during the vegetation growing season may increase by 5% to 10%. In arid and semi-arid regions, for every 10% increase in precipitation, FVC may increase by 8% to 15%. In terms of the impact of meteorological disasters on vegetation, research has mostly focused on the effects of drought (*Zhao, Gong & Liu, 2015*; *Orimoloye, Belle & Ololade, 2021*; *Ali et al., 2024*). For instance, under moderate drought conditions, the NPP of vegetation may decrease by 20% to 30% (*Li et al., 2019*). Under severe drought conditions, the decrease in NPP can exceed 40%. In areas where drought lasts for more than three months, the recovery time of FVC may be extended by 1 to 2 years (*Liu, Zhao & Wang, 2021*). In addition, the impact of meteorological disasters such as heavy rain on vegetation is also gradually attracting attention. The results show that extremely heavy rain can lead to a short-term decrease of 15% to 25% of the FVC, and will also have a lag effect on the long-term growth of vegetation, such as affecting the NPP of vegetation in the next growing season.

Vegetation is sensitive to changes in precipitation and air temperature (*Ahmad et al., 2023*). In karst regions, air temperature and precipitation are important meteorological factors affecting the growth of vegetation. It has been found that the response of NDVI to precipitation is significantly higher than that of air temperature (*Wei, Ren & Zhang, 2013*). In the context of climate change, the frequency and intensity of meteorological disasters have increased significantly (*Zhu & Xiong, 2018*; *Maryon, Zulaekhah & Nurendyastuti, 2023*; *Putri, 2021*). Rainfall, usually in the form of torrential rainfall disasters, is frequent, high and heavy. It is more destructive to soil and water conservation and vegetation growth in karst regions, but there are fewer studies on the impact of torrential rainfall on vegetation. Based on long-time series satellite remote sensing data and precipitation

observation data in karst areas of Guangxi, three vegetation remote sensing parameters of normalized difference vegetation index (NDVI), FVC and NPP and five rainfall levels (moderate rain, heavy rain, rainstorm, heavy rainstorm and extremely heavy rainstorm) are inverted and calculated. The temporal and spatial distribution characteristics of effects of different rainfall levels on vegetation are analyzed. At the same time, the impact of different rainfall levels on different forest species is studied, which provides a scientific basis for evaluating the impact of rainstorm and vegetation protection and restoration in karst areas.

# MATERIALS AND METHODS

## Overview of the study area

The Guangxi Zhuang Autonomous Region is located in the southern part of China, between 20°54′N and 26°23′N and 104°28′E and 112°04′E. This region is bordered by the Yunnan-Guizhou Plateau in the west, Guangdong Province in the east, Beibu Gulf in the south, and Hunan and Guizhou Provinces in the north. Its strategic location is significantly important. Guangxi has a complex and diverse terrain, mainly consisting of mountains and hills. Karst landforms are widely developed, accounting for 37.8% of the total land of the region. The climate is subtropical monsoon, characterized by warm and humid conditions, with abundant rainfall and distinct seasons. Annual average temperature ranges from 16 °C to 23 °C, and the annual precipitation varies from 1,200 to 2,000 millimeters. The vegetation types in Guangxi are rich and varied. Shrublands are the most widely distributed, accounting for 63.40% of the total vegetation area of the region, followed by broadleaf forests, accounting for 17.20%. Bamboo forests are the least, only accounting for 0.88%. In addition, Guangxi has a relatively high forest coverage rate, which reached 62.5% in 2023, ranking the top in the country. Distribution of the forests is shown in Fig. 1.

## Data

Daily temperature and precipitation date of 69 meteorological stations from 1961-2020 provided by the Guangxi Meteorological Information Center are used to calculate rainstorm disaster indicators (Table 1).

Satellite remote sensing data came from the Moderate Resolution Imaging Spectroradiometer (MODIS) product MOD13Q1 (MODIS/Terra Vegetation Indices 16-Day L3 Global 250 m SIN Grid), provided by the National Aeronautics and Space Administration (NASA). This dataset features a spatial resolution of 250 m and a temporal resolution of 16 d, providing high-quality synthesized time-series data. The data used corresponds to version V006 and covers the period from 2000 to 2021. The MOD13Q1 remote sensing dataset of Guangxi Karst region was preprocessed through a series of steps, including band extraction, mosaicking, projection transformation, region extraction, and data format conversion. This process generated a high-quality NDVI dataset, which is synthesized on a monthly basis to create a monthly NDVI dataset. Annual average NDVI values are calculated from this dataset.

Geographic information data includes digital elevation model (DEM) data, administrative boundaries of cities and counties, latitude and longitude coordinates of
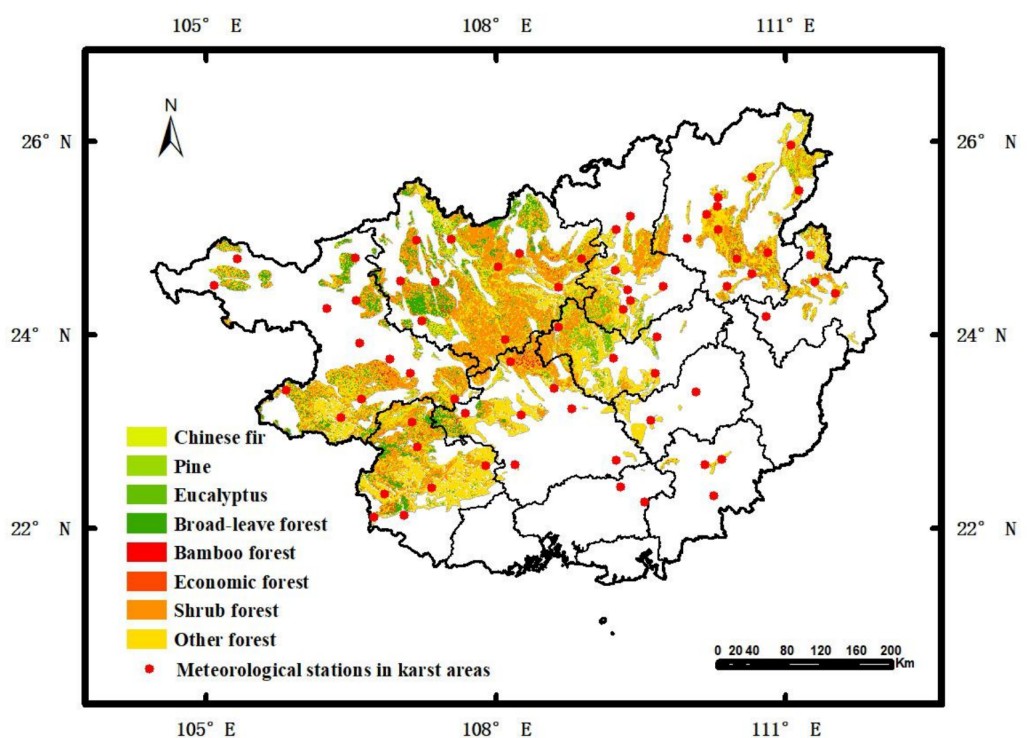

**Figure 1** Distribution of forest vegetation types and meteorological stations in the study area.

**Table 1 Indicators of rainstorm levels.**

| Rainfall | Definition | Unit |
|---|---|---|
| Extremely heavy rainstorm | Cumulative rainfall of daily precipitation (from 20:00 to next 20:00) ≥250 mm in a certain period | mm |
| Heavy rainstorm | Cumulative rainfall of daily precipitation (from 20:00 to next 20:00) ranging from 100 to 249.9 mm in a certain period | mm |
| Rainstorm | Cumulative rainfall of daily precipitation (from 20:00 to next 20:00) ranging from 50 to 99.9 mm in a certain period | mm |
| Heavy rain | Cumulative rainfall of daily precipitation (from 20:00 to next 20:00) ranging from 25 to 49.9 mm in a certain period | mm |
| Moderate rain | Cumulative rainfall of daily precipitation (from 20:00 to next 20:00) ranging from 10 to 24.9 mm in a certain period | mm |

meteorological stations, vector boundaries of karst areas, and d distribution data of forest species in Guangxi karst areas.

## Methods

Using terrestrial ecosystem carbon budget (TEC) models, the net primary productivity (NPP) can be calculated (*Xu, Hu & Wang, 2012*) as follows:

$$NPP_{ij} = GPP_{ij} - R_{ij} \tag{1}$$

$$GPP_{ij} = \varepsilon_{ij} \times FPAR \times PAR_{ij} \tag{2}$$

$$NPP_i = \sum_{j=1}^{n} NPP_{ij} \tag{3}$$

where $NPP_{ij}$, $GPP_{ij}$ and $R_{ij}$ (gC m$^{-2}$) are respectively the net primary productivity, total primary productivity and respiratory consumption of vegetation in the $j^{th}$ month of the $i^{th}$ year; $\varepsilon_{ij}$(gC MJ$^{-1}$) is the actual utilization rate of light energy in the $j^{th}$ month of the $i^{th}$ year, reflecting the influence of temperature, water and other factors on photosynthesis; FPAR is the proportion of photosynthetic active radiation absorbed by vegetation; $PAR_{ij}$ (MJ m$^{-2}$) is the incident photosynthetic active radiation in the $j^{th}$ month of the $i=^{th}$ year; $NPP_i$ (gC m$^{-2}$) is the net primary productivity of vegetation in the $i^{th}$ year, and $n$ is the total number of months in a year, $n = 12$.

Based on the image element linear decomposition model, vegetation cover is estimated using the NDVI and the pixel dichotomy method. That is, the NDVI value of each pixel can be expressed as a combination of contributions from two components: vegetation cover and non-vegetation cover. This relationship is quantified through the transformation of the vegetation cover fraction. FVC can be calculated as follows (*Liu, Liu & Liu, 2010*):

$$FVC_{ij} = \left(NDVI_{ij} - NDVI_s\right) / \left(NDVI_v - NDVI_s\right) \tag{4}$$

$$FVC_i = \frac{1}{n} \sum_{j=1}^{n} FVC_{ij} \tag{5}$$

where, $FVC_{ij}$(%) is the vegetation cover in the $j^{th}$ month of the $i^{th}$ year; $NDVI_{ij}$ is the NDVI in the $j^{th}$ month of the $i^{th}$ year; $NDVI_s$ and $NDVI_v$ are the NDVI of full soil cover and meta-vegetation full coverage, respectively, $NDVI_s = 0.05$ and $NDVI_v = 0.95$ based on the characteristics of vegetation in China; $FVC_i$ is the vegetation cover in the $i^{th}$ year. When $NDVI < 0.05$, the vegetation cover is negative, and there is no vegetation in the area.

The Pearson's correlation analysis is used to examine the relationship between remotely sensed indicators (such as FVC and NPP) for different vegetation types under different rainfall conditions (*Xu, Hu & Wang, 2012*). The calculation formula is as follows:

$$R = \frac{\sum_{i=1}^{n} (x_i - \overline{x})(y_i - \overline{y})}{\sqrt{\sum_{i=1}^{n} (x_i - \overline{x})^2 \sum_{i=1}^{n} (y_i - \overline{y})^2}} \tag{6}$$

where, $R$ is the correlation coefficient of variables $x$ and $y$; $x_i$ is the vegetation remote sensing parameters in the $i^{th}$ year; $\overline{x}$ is the mean of multi-year vegetation remote sensing parameters; $y_i$ is the rainfall in the $i^{th}$ year, and $\overline{y}$ is the mean of multi-year rainfall. The value range of correlation coefficient $R$ is $[-1, 1]$. The larger $R$ is, the stronger the correlation between the variables is. The significance test is conducted by $t$ statistic.

## RESULTS AND DISCUSSIONS

### Responses of vegetation remote sensing parameters to different rainfall levels

#### Fractional vegetation cover (FVC)

The average absolute values of the correlation coefficients between FVC and the different rainfall levels (extremely heavy rainstorm, heavy rainstorm, rainstorm, heavy rain and moderate rain) were 0.24, 0.19, 0.17, 0.17 and 0.19, respectively. These results indicate a weak correlation. The negative correlation areas accounted for 63.0%, 61.3%, 39.8%, 41.1% and 44.50%, respectively. There was a significant negative correlation in 23.3%, 1.58%, 0.61%, 1.16% and 1.00% of the areas ($p < 0.05$). FVC was mostly negatively correlated with extremely heavy rainstorms and heavy rainstorm, positively and negatively correlated with rainstorm, and positively correlated with heavy rain and moderate rain. The spatial differences in the correlation between FVC and different rainfall levels are significant (Fig. 2). The spatial distribution characteristics of FVC negatively correlated with rainfall indicates that: the high-value areas of the negative correlation between FVC and extremely heavy rainstorm were mainly located in the southwest (a). The negative correlation areas of heavy rainstorm were concentrated in the central region (b). The negative correlation areas of rainstorm and heavy rain were similar, mainly located in the southwest, north, and some parts of the northeast. The distribution was scattered, and the range of heavy rain was slightly larger than that of rainstorm (c and d). Moderate rain was mainly distributed in some parts of the north, with a relatively dispersed pattern (e). Comparing the correlation of FVC with different rainfall levels, FVC has the strongest correlation with extremely heavy rain, and the negative impact of extremely heavy rain on the vegetation is the most significant. The negative impact areas of extremely heavy rain and heavy rain are extensive, which further emphasizes the potential threat of extreme rainfall events to the vegetation ecosystem. From torrential rain to heavy rain, and then to extremely heavy rain, the correlation between FVC and them has changed significantly. Even the positive and negative correlation has changed in some areas. This change in correlation may be related to the factors such as increased rainfall intensity, leading to changes in soil moisture saturation, surface runoff, and physiological response of vegetation. As the rainfall level increases, soil moisture may reach saturation and surface runoff increase, putting greater pressure on vegetation and causing changes in correlation. In addition, the physiological adaptation mechanism and stress resistance of vegetation may also show different response characteristics under different rainfall intensities, thereby affecting the correlation between FVC and rainfall.

This study reveals the complex correlation between the vegetation cover fraction and different rainfall levels, as well as their spatial distribution characteristics in this area, providing an important scientific basis for a comprehensive understanding of the impact of rainfall on the vegetation ecosystem. The results cannot only help us understand the interaction mechanism between vegetation and rainfall, but also provide a useful reference for formulating targeted vegetation protection and restoration measures, as well as responding to the impact of climate change on ecosystems.

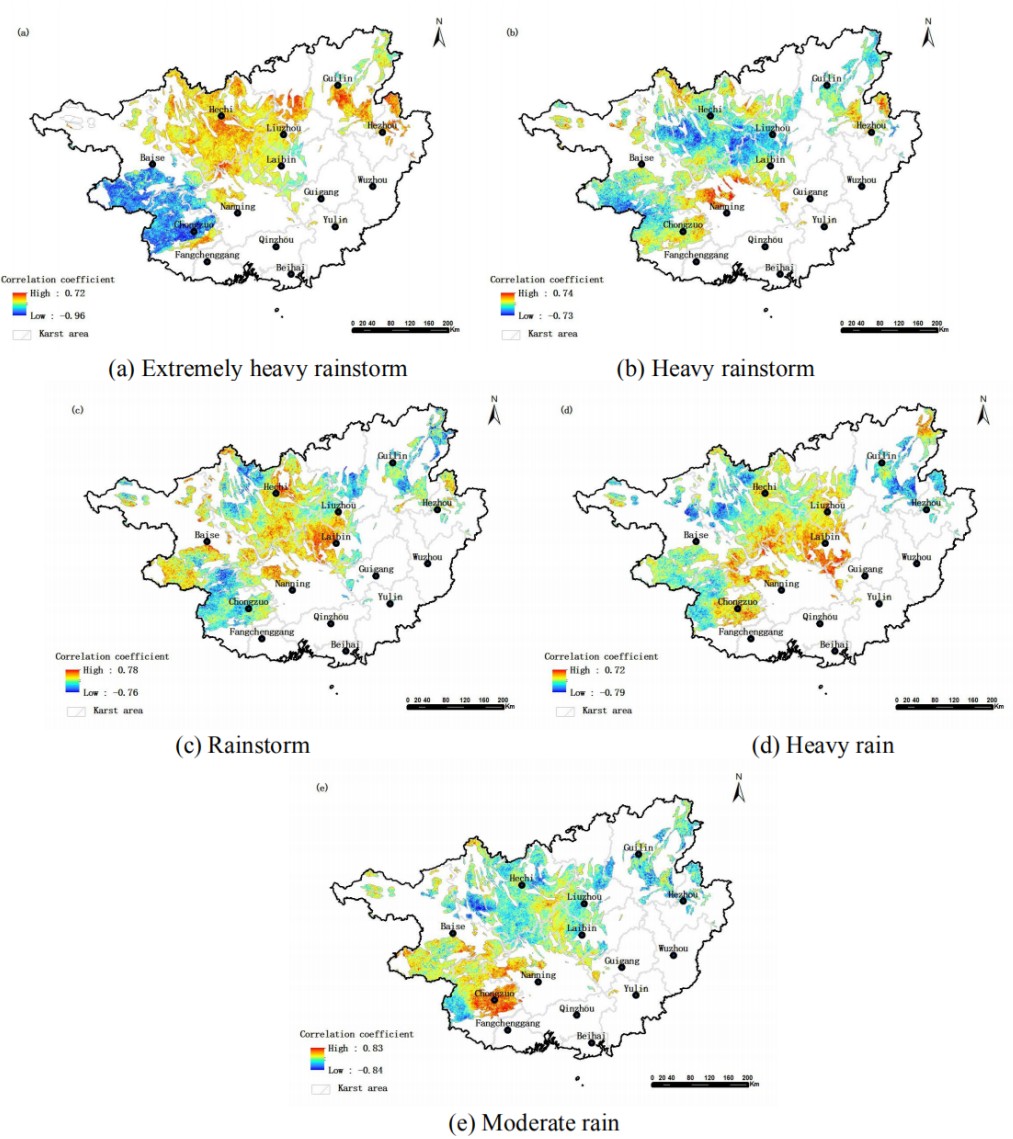

**Figure 2   Spatial distribution of correlation between FVC and different rainfall levels in karst areas of Guangxi.**

### Net primary productivity (NPP)

In this area, the absolute values of correlation coefficients between vegetation NPP and different rainfall levels (extremely heavy rainstorm, heavy rainstorm, rainstorm, heavy rain and moderate rain) were 0.23, 0.19, 0.22, 0.21 and 0.27, respectively. The values indicate a weak correlation. The proportion of areas with a negative correlation was 40.7%, 21.1%, 7.45%, 7.28% and 1.69%, respectively. Negative correlation was significant ($p < 0.05$) in 0.44%, 0.29%, 0.01%, 0% and 0% of the areas. NPP was mainly positively correlated with different rainfall levels. However, the negative correlation areas of extremely heavy rainstorms were relatively wide, with the proportion smaller than 50%. It is worth noting

that there were significant spatial differences in the correlation between NPP and different rainfall levels (Fig. 3). The high-value areas of negative correlation between NPP and extremely heavy rainstorm were located in the central and north-central parts of this area (a). Heavy rainstorm had a relatively concentrated distribution in some parts of the south-central and northeastern regions (b). Rainstorm was found in a few areas of the south-central part (c). Heavy rain was distributed in the east-central and northeastern parts of the area (d). The areas with a negative correlation for moderate rain were scattered and relatively few, appearing sporadically in the north, southwest and east (e). After conducting an in-depth analysis of the correlation between Net Primary Productivity (NPP) and different rainfall levels, there was a significant phenomenon: the negative correlation between NPP and extremely heavy rain was is the most pronounced. This finding indicates that extremely heavy rain, as an extreme rainfall event, has a relatively strong inhibitory effect on the net primary productivity of vegetation. Extremely heavy rain is typically characterized by high-intensity precipitation. The large amount of rainfall in a short period can lead to soil waterlogging, decreased soil oxygen content, and deteriorated light conditions. All of them have an adverse effect on the photosynthesis and growth processes of vegetation, thereby reducing its NPP. The spatial distribution characteristics of the negative impact of different rainfall levels on vegetation NPP reveal that the central region is a concentrated area where vegetation NPP is negatively affected by different rainfall levels. This area may have unique combinations of topography, soil texture, and vegetation types, making its vegetation NPP more susceptible to negative impacts under different rainfall levels.

Through the analysis of the correlation between NPP and different rainfall levels, as well as the study of the central region where vegetation NPP is negatively affected by rainfall, we not only reveal the significant impact of extreme rainfall events on vegetation NPP, but also provide important references and insights for the protection and sustainable development of the vegetation ecosystem in the central region.

## Responses of various forest species to different levels of rainfall

There are significant differences in the response of FVC to different rainfall levels in different forest vegetation types (Fig. 4). As the rainfall levels increased, the negative correlation between FVC and rainfall also increased. This was particularly evident when rainfall levels changed from heavy rainstorms to extremely heavy rainstorms, where the correlation between FVC and rainfall increased significantly. However, the impact of different rainfall levels on NDVI varies depending on forest vegetation. For extremely heavy rainstorm, broad-leaved forest ($-0.42$) and bamboo forest ($-0.40$) have the most significant impact, while eucalyptus ($-0.26$) and Chinese fir ($-0.27$) have relatively small impact. For heavy rainstorm, except for bamboo forest ($-0.13$), the impact on most forest vegetation types was not significant. In the case of rainstorm, Chinese fir ($-0.20$) showed a notable response, while eucalyptus ($-0.12$) was less affected. For heavy rain and moderate rain, the responses

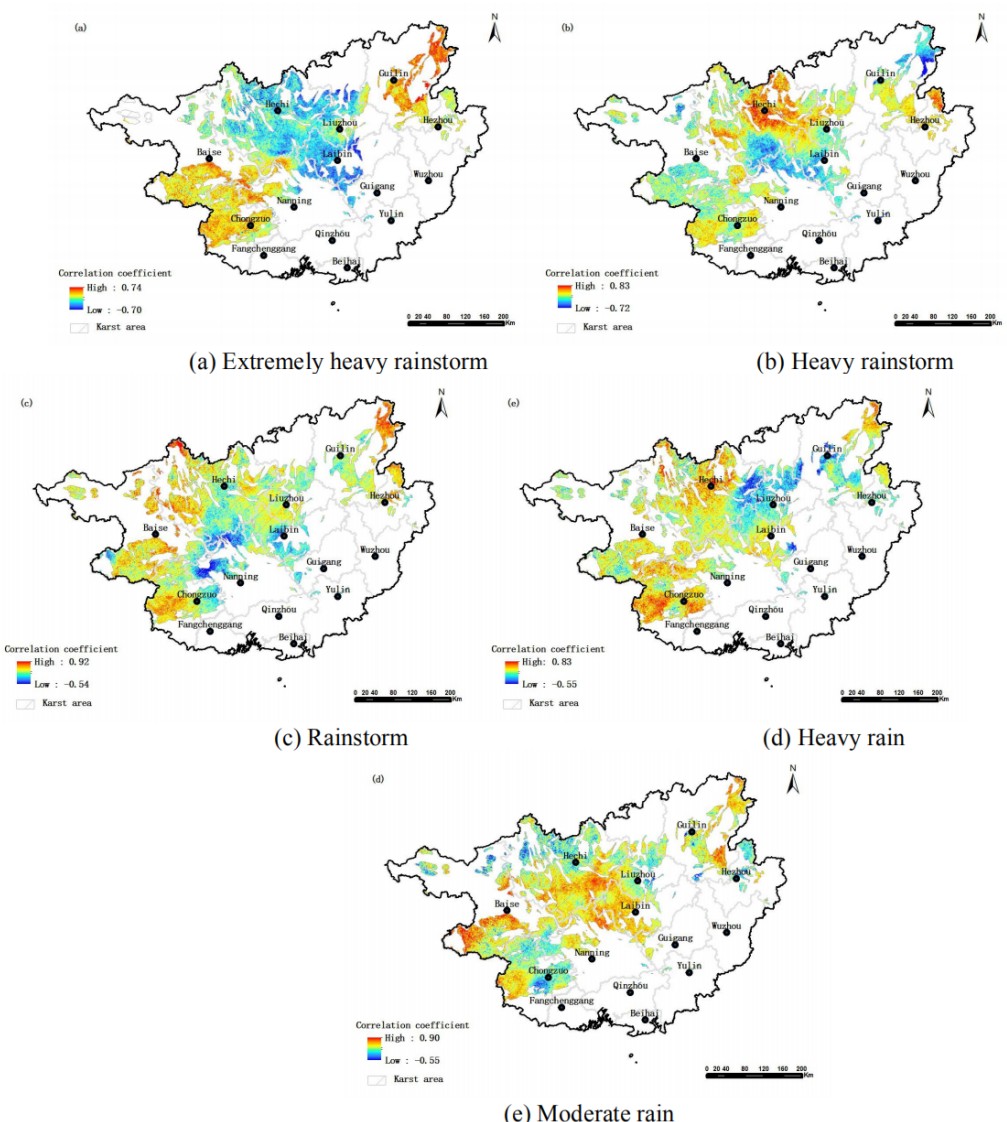

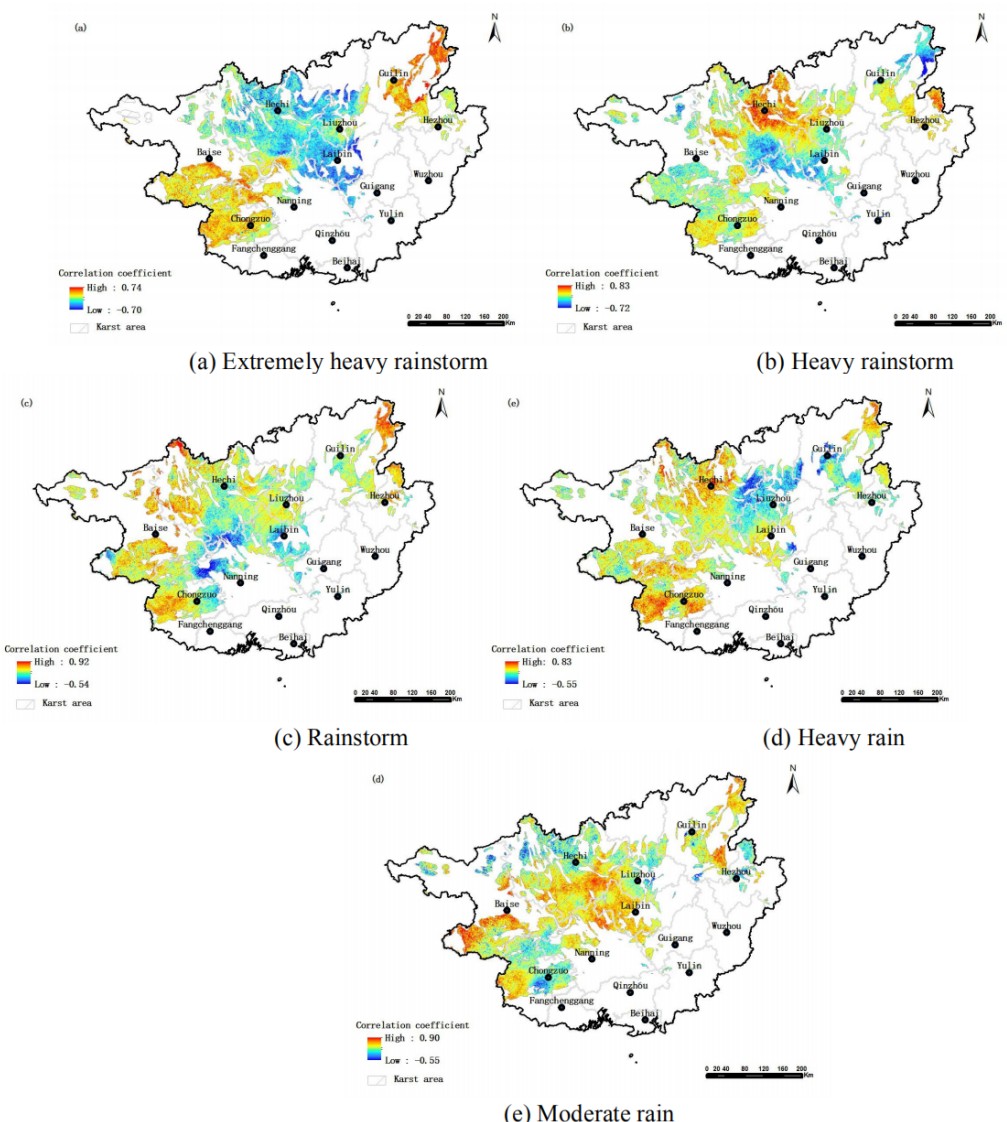

**Figure 3 Spatial distribution of correlation between vegetation NPP and different rainfall levels in karst areas of Guangxi.**

of forest types were consistent: economic forest (−0.19) and broad-leaved forest (−0.18) had strong impact, while eucalyptus (−0.11, and −0.12) showed minimal impact.

There are significant differences in the responses of NPP to rainfall levels in different forest vegetation types. Overall, as the rainfall levels increased, the negative correlation between NPP and rainfall increased. This trend was particularly pronounced when rainfall levels changed from heavy rainstorms to extremely heavy rainstorms, where the correlation between NPP and rainfall increased significantly. However, the effects of different rainfall levels on NPP varied depending on vegetation types. For extremely heavy rainstorms, the impact on eucalyptus forests (−0.21) were notable, while the effects on economic forests and broad-leaved forests (−0.13) were relatively small. Heavy rainstorms had

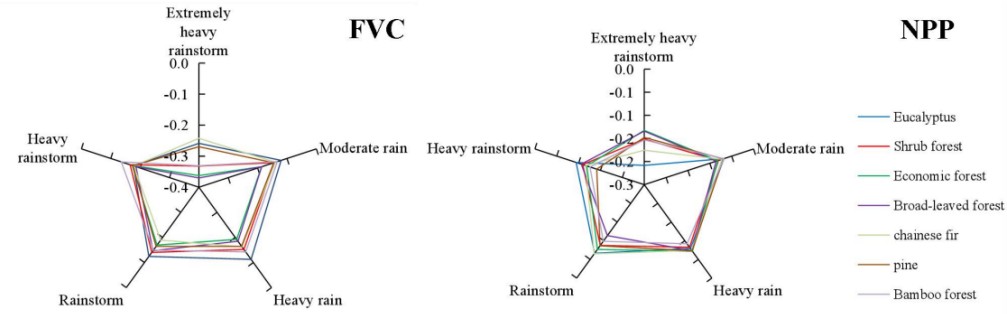

**Figure 4** **Negative correlation between various forest vegetation types and different rainfall levels in the study area.**

a more pronounced impact on pine forests (−0.14) and a less impact on eucalyptus forests (−0.09). In the case of rainstorms, broad-leaved forest (−0.11) showed a stronger response, while eucalyptus forests (−0.07) were less affected. The sensitivity of different forest vegetation was consistent for heavy rain and moderate rain.

The comparison of different remote sensing parameters of vegetation shows that there was a weak correlation between varying rainfalls levels and three vegetation remote sensing parameters. The negative correlation was the highest with NDVI, followed by FVC, and then NPP. Indifferent rainfalls levels, the negative effects of extremely heavy rainstorms were the most pronounced, followed by heavy rainstorms. In contrast, the effects of rainstorms, heavy rain, and moderate rain were weak and showed little difference. In different forest vegetation types, the negative effects of varying rainfall levels on NDVI and FVC were more pronounced in economic forest and broad-leaved forest, while the eucalyptus forest was small. However, the negative effects on NPP showed minimal differences in the forest vegetation types.

There was a certain difference in the negative impact areas of different rainfall levels on FVC of different forest vegetation types (Fig. 5). As the rainfalls level increased from rainstorm to heavy rainstorm, the proportion of negatively impact areas for different forest vegetation types increases. However, in other rainfall transition periods such as moderate rain to heavy rain, heavy rain to rainstorm and heavy rainstorm to extremely heavy rainstorm, this trend is not significant. Due to different rainfalls levels, the average negative impact area in all forest vegetation types ranged from 45%–61%. In terms of severity, the negative impact ranked as: heavy rainstorm > extremely heavy rainstorm > moderate rain > heavy rain > rainstorm. During extremely heavy rainstorm, the negative impact of eucalyptus (69%) and broad-leaved forests (66%) was greater than Chinese fir forest (48%). For heavy rainstorms, broad-leaved forest (66%) experienced a higher impact than eucalyptus (57%) and bamboo forests (58%). Under rainstorm conditions, Chinese fir forest (61%) showed a greater negative impact than bamboo forests (26%). In the case of heavy rain, economic forests (71%) were more affected than eucalyptus forests

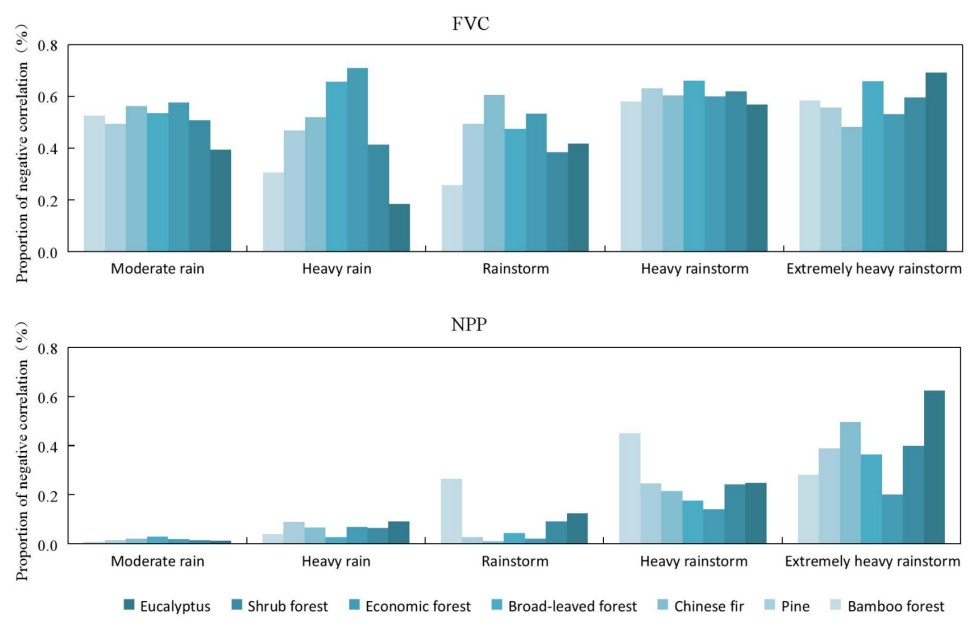

**Figure 5** Proportion of the negative correlation between different forest vegetation types and different rainfall levels in the study area.

(18%). Finally, under moderate rain, both economic (58%) and Chinese fir forests (56%) experienced a greater impact than eucalyptus forests (39%).

The negative impact of different rainfall levels on NPP varied in forest vegetation types. In general, as the rainfall levels increased from moderate rain to heavy rainstorm, the proportion of negative impacted area of most forest vegetation types increased significantly. However, when the rainfall changes from heavy rainfall to rainstorm, the proportion of areas negatively affected by economy, Chinese fir and pine forests will decrease. On average, the negative impact areas of all forest vegetation types ranged from 2%–39%, and ranked as: extremely heavy rainstorm > heavy rainstorm > rainstorm > heavy rain > moderate rain. For extremely heavy rainstorms, the negative impact area of eucalyptus (62%) was greater than that of economic forest (20%). Similarly, for heavy rainstorms, the impact area of bamboo forest (45%) was greater than that of broad-leaved forest (17%) and economic forest (14%). For rainstorm, the negative impact area of bamboo forests (26%) was greater than that of pine forests (3%), economic forest (2%) and Chinese fir forest (1%). For heavy and moderate rain, the negative impact areas in forest vegetation types were relatively small, ranging from 3% to 9% for heavy rain and from 1% to 3% for moderate rain.

Comparing different remote sensing vegetation parameters, it was found that there were significant differences in the proportion of negative impact areas of remote sensing for different rainfall levels and three vegetation covers. The proportion of negative impact areas of FVC was 53%, while the average of NPP was 16%. For FVC, heavy rainfall had the largest proportion of negative impacted areas, followed by moderate rainfall. The proportions of heavy and moderate rainfall were the smallest and relatively similar. In contrast, for NPP,

rainfall was proportional to the size of impact area. Regarding forests vegetation types, the proportion of negative impact areas of FVC was higher in economic and broad-leaved forests, but lower in eucalyptus and bamboo forests. Conversely, for NPP, the proportion of eucalyptus and bamboo forests was high, while that of economic and broad-leaved forests is low.

**Discussion**

Areas with sparse vegetation have poor water and soil retention capacity, making them more susceptible to soil and water loss during heavy rainfall events, which may further hinder vegetation growth (*Chen et al., 2015*; *Block & Richter, 2000*). In this study, it was found that vegetation in the northeast, northwest and southeast areas had a significant negative correlation with moderate rain and heavy rain. This suggests that in the ecologically sensitive and fragile karst areas, moderate and heavy rain may also have a notable impact on vegetation, in addition to the effects of rainfall levels such as rainstorms. Rainfall in karst areas of Guangxi is mainly concentrated in spring and summer. The spatial distribution of high-value areas for annual extremely heavy rainfall, heavy rainfall, and storm rainfall shows great similarity (a, b, and c). The high-value areas for annual moderate rainfall and heavy rainfall also have significant similarity (d and e). The northeastern region (Guilin City) and the central-western region (Hechi City) of the study area were identified as the main rainstorm centers and areas with concentrated rainfall (Fig. 6) (*Huang, Lin & Gao, 2012*). The spatio-temporal distribution of areas where vegetation is highly sensitive to extremely heavy rainstorm, heavy rainstorm and rainstorms in this area is significantly from that of regions with frequent rainstorm (the center and southwest). This indicates that in addition to spatio-temporal differences in precipitation (*Sun et al., 2021*), other factors such as variations in bedrock exposure rate in karst areas (*Chen et al., 2018*) and the complex composition of vegetation types (*Pan, Sun & Wang, 2021*), are also key contributors to the pronounced spatio-temporal heterogeneity of vegetation responses to rainfalls in these regions.

**CONCLUSION**

The negative impact of extreme rainfall on different remote sensing vegetation indices (vegetation coverage, FVC, and net primary productivity) in karst areas of Guangxi is most pronounced. In the two remote sensing vegetation indices, the negative impact of different rainfall levels on FVC is significantly greater than that on NPP. Moreover, the influence of different rainfall levels on the two remote sensing vegetation indices shows distinct spatial differences and varies greatly for different tree species. Overall, the negative impact of economic forests and broadleaf forests is more significant. It should be noted that rainfall also has a lag effect on vegetation, which will be further explored in future research.

However, the results may be influenced by various factors such as "seasonal effect". Vegetation may be more sensitive to heavy rainfall during the growing season (such as spring and summer), as it is in a rapid growth stage at this time. Heavy rainfall may lead to a decrease in vegetation cover (FVC) and hinder NPP. In the non-growing season (such as winter), the physiological activities of vegetation are relatively slow, and the impact of heavy

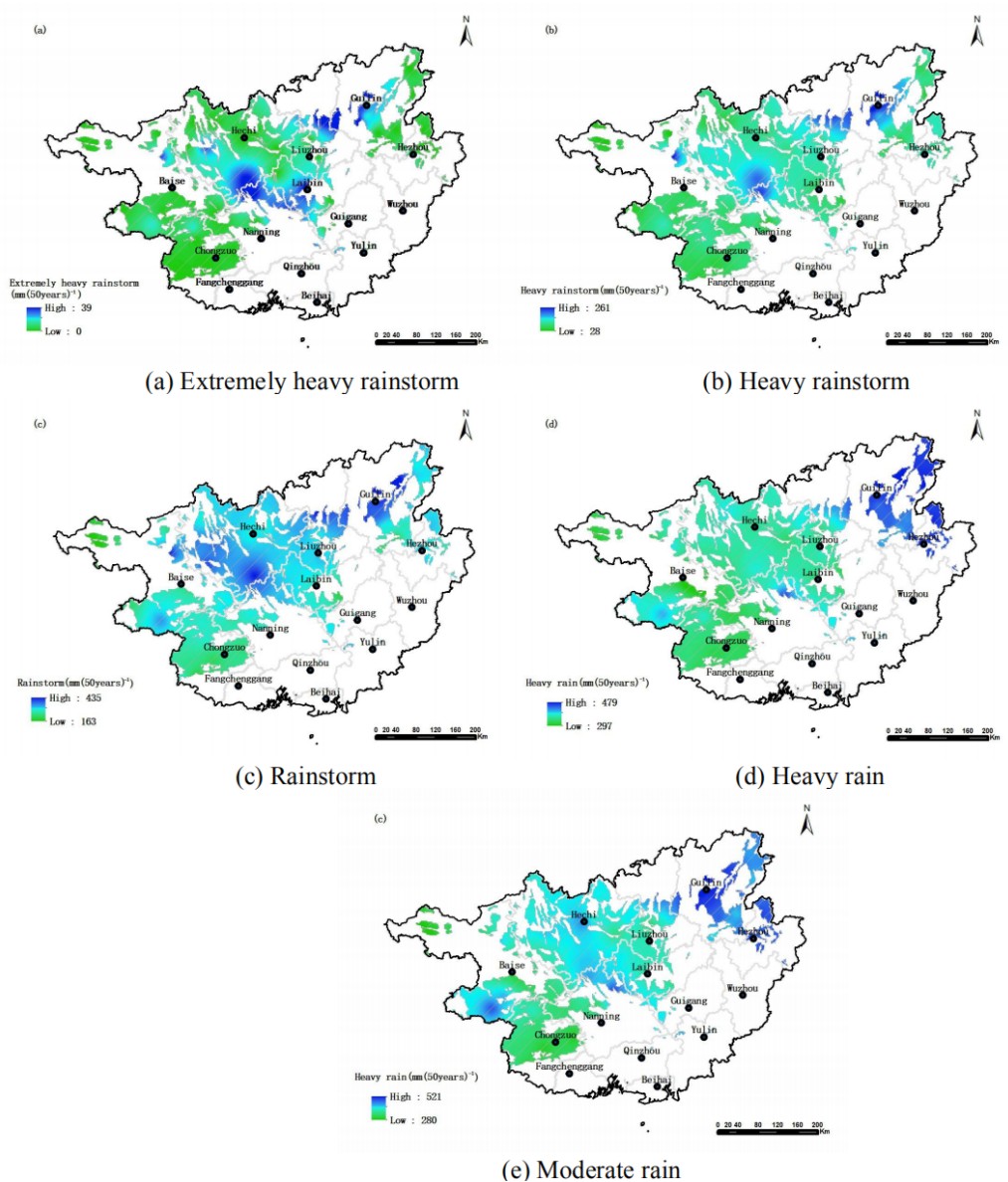

**Figure 6** **Spatial distribution of different rainfall levels in the study area.**

rainfall on it may not be significant. In addition, the intensity and duration of heavy rainfall in different seasons also vary, which may further affect the degree of damage caused by heavy rainfall to vegetation. It may also be affected by the "inter-annual effect". There are significant changes in climatic conditions (such as annual precipitation and temperature) in different years, which may affect the esistance of vegetation to heavy rainfall. In years with more precipitation, the soil moisture content is high, and heavy rainfall may lead to more serious soil erosion, thereby having a great negative impact on FVC and NPP. In years with less precipitation, vegetation may have some adaptability to heavy rainfall due to soil drought, and the impact of heavy rainfall on it may be relatively small. The terrain

in karst region of Guangxi is complex, and the impact of slope and aspect on vegetation during heavy rainfall should not be ignored.

Given the potential impact of the above factors, future research should further consider seasonal effects, inter-annual effects, and other related factors to more comprehensively assess the impact of heavy rainfall on vegetation in karst regions of Guangxi. Meanwhile, due to regional and tree species differences, targeted vegetation protection measures are of great significance, especially in areas with frequent extreme rainfall, economic forests, and broad-leaved forests. Efforts should be made to strengthen vegetation protection and ecological restoration, improve vegetation resistance to heavy rainfall, and enhance the stability of ecosystems.

### Funding
This work was supported by the Key Research and Development Program of Guangxi (Guike AB20159022, Guike AB23026052, Guike AB21238010). The funders had no role in study design, data collection and analysis, decision to publish, or preparation of the manuscript.

### Grant Disclosures
The following grant information was disclosed by the authors:
Key Research and Development Program of Guangxi: Guike AB20159022, Guike AB23026052, Guike AB21238010.

### Competing Interests
The authors declare there are no competing interests. Mingzhi Li is employed by Baise Meteorological Bureau.

### Author Contributions
- Mingzhi Li conceived and designed the experiments, prepared figures and/or tables, and approved the final draft.
- Ying Xie conceived and designed the experiments, prepared figures and/or tables, authored or reviewed drafts of the article, and approved the final draft.
- Yanli Chen performed the experiments, prepared figures and/or tables, authored or reviewed drafts of the article, and approved the final draft.
- Yue Zhang performed the experiments, prepared figures and/or tables, and approved the final draft.
- Weihua Mo analyzed the data, prepared figures and/or tables, and approved the final draft.

### Data Availability
The raw measurements are available in the Supplementary Files.

## Supplemental Information

Supplemental information for this article can be found online at http://dx.doi.org/10.7717/peerj.19565#supplemental-information.

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
