# Peer review of "Impact of precipitation levels on vegetation in ecologically fragile karst areas in the Guangxi (China) karst region"

_PeerJ, doi:10.7717/peerj.19565_

## Round 0.1 · original submission · Major Revisions

Despite being composed in English, the document requires more clarity and coherence for improved fluidity. The text lacks clarity, rendering it challenging to comprehend. The text lacks clarity, rendering it challenging to comprehend. Therefore, I recommend that you examine it, as the corrections are included in the paper.

·

Basic reporting

Although the document is written in English, it needs clarity and coherence to make it fluid. It is not clear and therefore difficult to read. It is not clear and consequently difficult to read. Hence I suggest you review it and the corrections are in the document.

I believe the document requires significant revisions. The summary lacks essential information, including the forest vegetation types and precipitation levels assessed, which should be explicitly stated. Throughout the document, I have provided considerations and suggestions for changes that could enhance its clarity and comprehensiveness.

Experimental design

The methods section lacks information, or perhaps how it is written makes it unclear and difficult to understand. Suggestions have been made in the text to improve clarity and to include important details that are currently missing.

Validity of the findings

Throughout the document, significant correlation values (r and P) are not presented, and the types of forest vegetation analyzed are not described.
Additionally, results are mixed with the discussion, making it necessary to reorganize these sections for better clarity, consistency, and grammatical accuracy.

Additional comments

Throughout the document, significant correlation values (r and P) are not presented, and the types of forest vegetation analyzed are not described.

There is repetition and redundancy in different sections of the document.


Tables and Figures
Table 1
If the units have already been provided in the definitions, why repeat a column with the same units?
Figures
Figures 2, 3, and 6 reference indices (a–e), but these indices are not present in figures. Clarification is needed on what they refer to.
The numbering of figures does not match their references in the text. Additionally, there are more figures than those listed in the figure index.

Reviewer 2 ·

Basic reporting

In Language,
1. The author is accustomed to using long sentences, which exacerbates the difficulty of reading.

Experimental design

In Materials and methods,
1) what is the standard for the defined rainstorm levels?
2) The author needs to provide an accurate basis for the data source and its license acquisition.
3) Some method references should be added.
4) The statistical analysis should be expanded.

Validity of the findings

In Results,
1) The author needs to consider why the spatial distribution of correlation between FVC or NPP and different rainfall levels is established separately in Fig. 2 or Fig. 3, and what is the overall situation of the region?
2) The author needs to consider why the spatial distribution results of different rainfall levels are presented in Fig. 6, and what is the logic behind it? What are the spatial distribution of FVC and NPP?
3) The author needs to explain in depth whether the results of this study will be affected by the "season effect" or "yearly effect" or other factors, this seems to have not been mentioned.

Additional comments

In Introduction,
There are some major concerns as follows,
1) The author needs to strengthen the proposed hypothesis, mainly to provide a detailed explanation of rainfall patterns and vegetation status in Karst areas, especially these concerned parameters in the manuscript, such as climate factors.
2) The author needs to highlight the significance and advantages of this research topic or purpose compared to previous similar studies.
3) The author needs to consider the impact of climate factors, including rainfall, and other factors, such as altitude, topography, vegetation types and soil characteristics, on the research results.

Reviewer 3 ·

Basic reporting

.

Experimental design

.

Validity of the findings

.

Additional comments

Authors reported response of vegetation to different levels of rainfall in ecologically fragile areas of Guangxi Karst. The topic is of relevance and falls within the interests of the journal. The techniques used are adequate and the results are effectively presented and discussed. Thus I suggest considering the manuscript for publication if the authors will provide a minor revision addressing the issues listed below.
1. Abstract section, some important values are suggested to be added.
2. Figure 5, the colors are difficult for readers to distinguish. Authors should revise it.
3. Conclusion section, Merge two paragraphs.
4. The results are suggested to be compared with previous reports.

---

## Round 0.2 · accepted · Accept

This revised version is suitable for publication in PeerJ.

Reviewer 2 ·

Basic reporting

Most of the reviewer's comments and suggestions have been taken on consideration by the authors of this paper and also the questions have been clearly answered.

Experimental design

-

Validity of the findings

-

Additional comments

-

Reviewer 3 ·

Basic reporting

The corrections have been done. This version is acceptable for publication.

Experimental design

Good.

Validity of the findings

Good.

Additional comments

No.